# Pyrolysis Kinetics of *Byrsonima crassifolia* Stone as Agro-Industrial Waste through Isoconversional Models

**DOI:** 10.3390/molecules28020544

**Published:** 2023-01-05

**Authors:** Jonathan M. Sanchez-Silva, Raúl Ocampo-Pérez, Erika Padilla-Ortega, Diakaridia Sangaré, Miguel A. Escobedo-Bretado, Jorge L. Domínguez-Arvizu, Blanca C. Hernández-Majalca, Jesús M. Salinas-Gutiérrez, Alejandro López-Ortiz, Virginia Collins-Martínez

**Affiliations:** 1Centro de Investigación y Estudios de Posgrado, Facultad de Ciencias Químicas, Universidad Autónoma de San Luis Potosí, San Luis Potosí 78260, Mexico; 2Institut Universitaire de Technologie, Université d’Orléans, 16 Rue d’Issoudun, BP16724, CEDEX 2, 45067 Orléans, France; 3Facultad de Ciencias Químicas, Universidad Juárez del Estado de Durango, Av. Artículo 123 s/n Fracc, Filadelfia, Gómez Palacio 35010, Mexico; 4Centro de Investigación en Materiales Avanzados, Departamento de Ingeniería y Química de Materiales, S.C., Miguel de Cervantes 120, Chihuahua 31136, Mexico

**Keywords:** pyrolysis kinetic triplet, *Byrsonima crassifolia*, lignocellulosic biomass, isoconversional methods, thermal decomposition

## Abstract

This study is aimed at the analysis of the pyrolysis kinetics of Nanche stone BSC (*Byrsonima crassifolia)* as an agro-industrial waste using non-isothermal thermogravimetric experiments by determination of triplet kinetics; apparent activation energy, pre-exponential factor, and reaction model, as well as thermodynamic parameters to gather the required fundamental information for the design, construction, and operation of a pilot-scale reactor for the pyrolysis this lignocellulosic residue. Results indicate a biomass of low moisture and ash content and a high volatile matter content (≥70%), making BCS a potential candidate for obtaining various bioenergy products. Average apparent activation energies obtained from different methods (KAS, FWO and SK) were consistent in value (~123.8 kJ/mol). The pre-exponential factor from the Kissinger method ranged from 10^5^ to 10^14^ min^−1^ for the highest pyrolytic activity stage, indicating a high-temperature reactive system. The thermodynamic parameters revealed a small difference between E_A_ and ∆H (5.2 kJ/mol), which favors the pyrolysis reaction and indicates the feasibility of the energetic process. According to the analysis of the reaction models (master plot method), the pyrolytic degradation was dominated by a decreasing reaction order as a function of the degree of conversion. Moreover, BCS has a relatively high calorific value (14.9 MJ/kg) and a relatively low average apparent activation energy (122.7 kJ/mol) from the Starink method, which makes this biomass very suitable to be exploited for value-added energy production.

## 1. Introduction

It has been established that climate change is the greatest environmental problem in the world today, as it also has a negative impact on the economy and the energy sector. This phenomenon is mainly caused by environmental pollution and the emission of greenhouse gases from the overexploitation of fossil fuels, so it is urgent to search for and develop alternative energy sources [1].

Among various renewable energy sources, biomass has gained great attention for its use as bioenergy, so it is considered the fourth most relevant energy source after coal, oil, and natural gas [2]. Biomass derived from lignocellulosic residues from different agro-industrial sectors is considered a sustainable and viable option for valorization through energy generation due to its high abundance, low cost, low greenhouse gas emissions, and carbon neutrality [3,4]. Moreover, agro-industrial residues contain three main components: hemicellulose (20–40%), cellulose (40–60%), and lignin (10–25%) [5], which favor their conversion into liquid, solid and gaseous products depending on the conversion process used (biochemical, thermochemical, and physicochemical) [6].

*Byrsonima crassifolia*, better known as Nanche, is a fruit species distributed in tropical and subtropical zones of México, central and south America [7]. The fruit is harvested from July to September and is a yellow-orange drupe with abundant fibrous pulp surrounding an endocarp, which is hard and woody and commonly called “stone” [8]. Recently the consumption of this fruit has increased, and this is due to its high content of antioxidants and unsaturated fatty acids [9,10]. In México during 2020, 9262 tons of fruit were produced, representing a planted area of 1752 hectares [11], and as the endocarp corresponds between 25–34% of the total weight of the fruit [12], it is considered an important agro-industrial waste, with the potential to be valorized through some thermochemical conversion process such as liquefaction, combustion, hydrothermal carbonization, gasification or pyrolysis, which involve the conversion of the main constituents of biomass into valuable energy products [13].

Thermochemical processes allow the valorization of agro-industrial wastes for the generation of gaseous, liquid, and solid fuels [14]. Among the various processes, pyrolysis is a renewable, economical, and efficient way to convert agro-industrial wastes [15]. However, understanding the factors involved in the process remains a challenge, primarily in the design, scale-up, and operation of pyrolysis. In this context, pyrolysis is defined as the thermal degradation of organic matter in the absence of oxygen from which products such as biochar, bio-oil, tar (aqueous solution of organic compounds), and gaseous products are obtained [16]. Pyrolysis occurs at temperatures between 300 and 600 °C in an inert atmosphere, which prevents combustion and gasification reactions from occurring. It is important to note that during pyrolysis, the products obtained are more valuable than the raw material [14].

The thermal degradation behavior of fuels, compounds, and biomass is generally studied using data obtained from thermogravimetric analysis (TGA) and differential thermal analysis (DTG). Experimental data obtained at different heating rates provide valuable information on the thermal behavior of biomass. In fact, this method has been generally used to study the characteristics, thermal behavior, and degradation kinetics of biomass due to its high accuracy and simplicity [16,17]. From this analysis, it is possible to evaluate the kinetic triplet consisting of the apparent activation energy (E_A_), pre-exponential factor (A_α_), and reaction mechanism or kinetic model, f(α) [6,18] and, in turn, estimate the thermodynamic properties of the process (∆H, ∆G, and ∆S). These parameters are essential for investigating and evaluating the energetic feasibility of a pyrolysis process [13,18].

In the literature, there are several studies of biomass pyrolysis using various agro-industrial wastes such as avocado stone, *Agave salmiana* bagasse, and cocoa shell by Sangaré et al. [18], apple pomace by Baray-Guerrero et al. [19], garlic husk by Singh et al. [20], peanut shells by Açıkalın [17], corn stalk by Cai et al. [15], wheat straw by Mani et al. 2010 [21] and waste sawdust by Mishra & Mohanty [5], using various kinetic methods. However, the reported results are different for each agro-industrial waste due to different physicochemical characteristics and pyrolytic operating conditions [22]. The kinetic analysis of biomass thermal degradation is generally based on the degradation rate equation developed by Friedman in 1964 [23], and the ICTAC (International Confederation for Thermal Analysis and Calorimetry) committee has evaluated integral isoconversional methods such as Kissinger–Akahira–Sunose (KAS), Flynn–Wall–Ozawa (FWO) and Starink (SK), and concluded that the SK method is the most valid and reliable integral isoconversional method for the estimation of apparent kinetics [24]. This method can provide effective and accurate activation energies for kinetic analysis of biomass pyrolysis [1,15,24]. Therefore, the physicochemical characterization composed of the proximal and compositional analysis of biomass, the kinetic study involving the kinetic triplet in combination with the calculation of the thermodynamic parameters; enthalpy (∆H), Gibbs free energy (∆G), and entropy (∆S) of the pyrolysis process, represent a set of information that is fundamental for the design, construction, and operation of a large-scale reactor for the pyrolysis of any lignocellulosic residue [16].

From the above, the objective of the present research is to perform the first study on the pyrolysis kinetics of Nanche (*Byrsonima crassifolia*) stone as biomass waste by evaluating its physicochemical characterization and the kinetic triplet; apparent activation energy, pre-exponential factor, and reaction model, and from these to estimate its thermodynamic parameters (∆H, ∆G, and ∆S). It is expected that from the results obtained in the present investigation, it will be possible to evaluate the potential of the Nanche stone to be exploited as a valuable bioenergy resource.

## 2. Results and Discussions

### 2.1. Biomass Physicochemical Characterization

Table 1 presents the results of the characterization of the Nanche stone. From these results, it is evident that there is a low ash content (2.2 wt %), a high volatile content (71.7 wt %), and a significant amount of fixed carbon (14.8 wt %), which is characteristic of lignocellulosic biomasses, making this residual biomass attractive for thermal degradation processes and biochar generation [25]. Regarding the mineral composition of the ash, the highest contributions come from alkali metals such as potassium (76.38 wt %), sodium (7.95 wt %), and calcium (6.66 wt %), this high composition is beneficial for the thermochemical conversion process. It has been shown that these metals could modify the crystalline structure of cellulose and further promote reactivity during pyrolysis [26]. The elemental analysis shows that the Nanche stone has a small amount of N (1.52 wt %) and no S, which is advantageous because it minimizes corrosion problems associated with the formation of acids in thermochemical processing equipment. It is also evident that the highest elemental amount corresponds to carbon, with 49.88 wt %, followed by oxygen, with 42.95 wt %. From the lignocellulosic composition, it is important to note the high content of cellulose (44.16 wt %) and lignin (34.67 wt %) which gives high rigidity and hardness to the biomass. Finally, the calorific power value (14.93 MJ/kg) is among values commonly found in different biomasses such as Rice husk (17.96 MJ/kg), Wheat Straw (15.29 MJ/kg), Sugar cane bagasse (18.56 MJ/kg), Banana trunk (13.41 MJ/kg), and Rice Straw (15.06 MJ/kg) [27], where biomass having higher volatile matter content and low moisture content has high calorific power value and better ignition and combustion rate [5].

### 2.2. Thermogravimetric Analysis (TGA & DTG)

#### 2.2.1. Particle Size Effect

The experimental TGA and calculated DTG curves obtained for the Nanche endocarp under a heating rate of 5 °C/min and variable particle size are presented in Figure 1. Here it can be observed that the devolatilization zone occurs from 250 °C to 450 °C. Furthermore, a decrease in particle size Dp < 600 µm causes a higher devolatilization of the biomass ~79.050%, i.e., a higher amount of volatile material is generated and thus a lower amount of biochar (Figure 1) [28]. From the DTG curves (Figure 1), it can be observed changes in hemicellulose degradation (shoulder at ~275 °C). This can be associated with a clogging effect between the cross-linked hemicellulose in the lignin, i.e., when the particle size is smaller, there is a larger exposure area for the decomposed hemicellulose to be released more easily into the inert atmosphere. In addition, it can be seen that an increase in particle size causes a shift in the temperature of the maximum peak in the DTG curve, as well as a difference in the decrease of the cellulose degradation peak between 340–367 °C, which may be caused by some heat and mass transfer problems [29]. For the analysis of the effect of heating rate, the particle size (850 µm < Dp_4_ < 1680 µm, mesh #20) was chosen because it presents the most consistent behavior. That is, this DTG curve presents well-defined characteristic peaks of hemicellulose, cellulose, and lignin, which will facilitate the complete analysis by means of the deconvolution technique that is described below.

#### 2.2.2. Heating Rates Effect

The TGA and DTG curves obtained from the Nanche endocarp by modifying the heating rate are presented in Figure 2. The thermal degradation is divided into three stages, as shown in the corresponding figure. The DTG curves initially indicate endothermic dehydration and moisture loss in the first stage (Drying stage, 30–150 °C) with a mass loss between 3.21 to 3.82%, which means that the biomass has less than 5% moisture, so it can be considered feasible for combustion. The decay in the degree of conversion during the second stage (active pyrolytic stage, 150–450 °C) is characteristic of lignocellulosic biomasses [30], and this stage has two zones, i.e., in a temperature range of 250 to 300 °C (Z_1_) corresponds to the degradation of hemicellulose. Subsequently, at a temperature between 320 °C and 380 °C (Z_2_) indicates the degradation of cellulose and part of lignin, and it is this stage that contributes most to mass loss (>60%). Finally, the third stage (passive pyrolytic stage, > 450 °C) corresponds to the degradation of lignin and the formation of charcoal. Yang et al. [31] report the temperature ranges for the degradation of hemicellulose, cellulose, and lignin as 220–315, 315–400, and 160–900 °C, respectively, and more specifically, it has been found that cellulose degrades between 277 and 427 °C, hemicellulose around 197 and 327 °C and lignin between 277 and 527 °C [32]. Furthermore, it is observed that when the heating rate is high, the degradation rate becomes slower due to the restriction of heat transfer between the particles, as opposed to a slow heating rate where the heat remains in the biomass for a longer time, thus resulting in an intense heat transfer between the particles favoring a higher degradation rate, allowing a greater amount of dehydration, depolymerization, carbonylation, carboxylation, and transglycosylation reactions [28], thus causing a decrease in the DTG_max_ value (%/°C) presented in Table 2. Concerning the fact that a reverse behavior is observed in the temperature range of 315–400 °C with respect to the hemicellulose decomposition, this can be explained by the behavior of the cellulose decomposition reported by Várhegyi et al. [33], where mass transfer problems caused by high heating rates can delay the decomposition process, and in the case of cellulose the presence of reaction products during its decomposition can initiate autocatalytic reactions and cellulose can be consumed below the maximum cellulose decomposition temperatures, thus causing a shift and decrease in the maximum peak cellulose degradation of DTG’s as in the present study. Finally, yields of 31.61, 29.84, and 30.45% were observed for heating rates of 5, 10, and 15 °C/min, respectively, at a temperature of 600 °C. Thus, Nanche endocarp can be considered a very good precursor for the generation of biochar and volatile compounds by slow pyrolysis [34].

Saffe et al. [35] have reported an estimation of hemicellulose, cellulose, and lignin contents of various agro-industrial wastes based on a deconvolution technique of the dα/dT vs. T profiles as observed in Figure 3. This technique is applied to clearly separate all the peaks corresponding to the degradation of the three natural polymers contained in the biomass, and their contents are determined with the size ratio of the areas under these peaks. The deconvolution of the dα/dT vs. T curves in the present study was carried out using Origin 9.0 software. Figure 3 shows the results obtained using this technique for the heating rate of β = 5 °C/min. A Gaussian equation was used to fit each peak as described by Perejón et al. [36] where a mathematical model was proposed, and its integration allowed to estimate the percentage of hemicellulose, cellulose, and lignin content corresponding to the Nanche stone.

According to the results of Figure 3, it can be observed that the deconvolution of the first peak, which corresponds to hemicellulose, starts at a temperature of ~204 °C and ends at around 380 °C, presenting a maximum peak at 285.8 °C. A second peak can also be seen here, which is related to cellulose and starts at a temperature of approximately 220 °C and ends at 440 °C, presenting a maximum peak at 339.6 °C. Finally, a third peak can be identified because of deconvolution that starts at approximately 240 °C and ends at 700 °C, presenting a maximum peak at 421.7 °C, which is related to the lignin content of the Nanche stone. These results agree well with what has been previously reported in the literature for this type of lignocellulosic biomass [37]. According to the results of Figure 3, it can be observed that due to the deconvolution, the corresponding percentage of area for each of the three peaks obtained corresponds to the theoretical percentage content of hemicellulose, cellulose, and lignin of 34%, 43%, and 23%, respectively. It is worth mentioning that these results agree significantly well with those obtained from the compositional analysis reported in Table 1.

### 2.3. Kinetic Triplet Analysis

#### 2.3.1. Apparent Activation Energy

Linear fit plots using the KAS, FWO, and SK isoconversional methods to determine the apparent activation energy are presented in Appendix A, found in the Appendix A. From these methods, the isoconversional lines are parallel for a specific degree of conversion (α) from 0.1 to 0.65, and when the degree of conversion is α ≥ 0.7 slight deviations occur, so it can be considered that the reactions from this degree of conversion and above become more complex [6]. Appendix A shows the apparent activation energy obtained from the various isoconversional methods. For the three methods KAS, FWO, and SK, there is a region where the apparent activation energy has a maximum value when α is at ~0.5 (334.4–352.2 °C) and which corresponds to the second zone of the active pyrolytic stage, i.e., the region where the peak of cellulose degradation occurs as shown in the DTG diagram (Figure 2). This is because a higher amount of energy is required to degrade the cellulose in the biomass. Furthermore, it was found that the apparent activation energy decreases when the degree of conversion increases from 0.5 to 0.85, indicating that the pyrolysis of BCS involves a complex set of reactions remaining very little amount of cellulose and hemicellulose, after a degree of conversion α > 0.5 [20] thus decreasing the E_A_. The mean values of the calculated apparent activation energy were: 122.3 kJ/mol (KAS), 126.3 kJ/mol (FWO), and 122.7 kJ/mol (SK) (Tabulated data can be found in the Appendix A). According to the ICTAC recommendations, the Starink method was chosen as the best model that represents the apparent activation energy during BCS pyrolysis. The results obtained with the SK model were in good agreement with a deviation of less than 6%, which validated the reliability of the calculations and confirmed the predictive power of the SK method, and the data obtained from this method was used to perform subsequent calculations in the present study.

Since the apparent activation energy is the minimum energy needed for a reaction to proceed, the higher the E_A_ values, the more difficult it will be to initiate the reactions involved in pyrolysis. El May et al. [38] reported the kinetic model of date stone and calculated an E_A_ value of 58.9 kJ/mol, which is a relatively low value compared with the present study (BCS) since date stone has a different nature with respect to Nanche stone. However, when comparing the apparent activation energy of the present study with respect to other stones of similar characteristics, such as olive and plum stones, the differences are rather small. Garcia et al. [39] report a value of 198.7 kJ/mol for olive stone pyrolysis, while Ceylan [37] presents an E_A_ value of 154.8 kJ/mol for plum stone pyrolysis. Damartzis et al. [40] studied the pyrolysis kinetics of cardoon (*Cynara cardunculus*) and found an E_A_ of 223.8 kJ/mol. Amutio et al. [41] studied the pyrolysis of pine wood residues and found that E_A_ ranged from 63 to 205.8 kJ/mol. In general, the E_A_ values calculated for Nanche stone are similar to those reported in previous studies on biomass residues. It is important to note that the apparent activation energy values determined for different degrees of conversion should not be considered as the actual values for a specific reaction step but, rather, as an apparent value representing the contributions of various parallel and competing reactions that contribute to the overall reaction rate [15,19].

Table 3 shows the values of the apparent activation energy obtained from the FWO model of various biomasses, their structural composition, and elemental ratios, as well as one of the most important thermal parameters of biomass and waste solid fuels, indicating their energy content, is the higher heating value (HHV). HHV features the maximum amount of energy potentially recoverable from solid fuels [42]. According to these values, it can be clearly seen that cellulose has a high apparent activation energy (166.42 kJ/mol). However, cellulose has a relatively intermediate higher heating value (15.10 MJ/kg). While cocoa husk has the highest HHV value (21.06 MJ/kg), and this can be correlated with the elemental components of the biomass, as shown in the Van-Krevelen diagram in Figure 4, where all the analyzed biomasses differ in their O/C and H/C ratios. However, when plotting the O/C ratio vs. HHV value, it is clearly shown that biomasses having a small O/C ratio value have the highest HHV value. In addition, it is known that there is a strong correlation between apparent activation energy and lignin content, which influences the apparent activation energy to be governed by the structural composition of biomass. When biomass has a high apparent activation energy, it is not necessarily beneficial, as it implies that not all the energy used for thermal degradation of the biomass in the pyrolysis process will be used to obtain energetically valuable solid, liquid, and gaseous products, e.g., the cocoa shell contains a high content of hemicellulose, which represents a high content of organic matter that is formed by relatively weak energetic bonds from unfolding and would presumably give rise to simpler combustible products (liquid and gaseous) than those where the cellulose and lignin content is higher [43]. For example, in the case of apple pomace and *Phyllanthus emblica* kernel, which have a higher cellulose and lignin content, a higher HHV accompanied by a moderate increase in apparent activation energy is reflected. This energy balance is the one presented by the BCS endocarp since it has a significant amount of cellulose accompanied by moderate amounts of both hemicellulose and lignin that represent a relatively high HHV value (17.92 MJ/kg) accompanied by a relatively low average apparent activation energy (122.7 kJ/mol) compared with other biomasses presented in Table 4, which makes this biomass very suitable to be exploited in pyrolytic transformations to obtain value-added energy.

#### 2.3.2. Pre-Exponential Factor (A_α_)

Values of the pre-exponential factor were calculated from the Kissinger method using the apparent activation energies of the SK method. The pre-exponential factor is an important parameter involved in pyrolysis and fundamental for pyrolysis reactor design and optimization [18,46]. Using Equation (13), the values of A_α_ as a function of the degree of conversion at a heating rate of 15 °C/min were determined and are presented in Table 4. The values of A_α_ involved in the pyrolysis of BCS presented variations in a range from 10^5^ to 10^14^ min^−1^ when the degree of conversion increases from 0.1 to 0.7. This is because in this range of conversion is where the highest pyrolytic activity caused by the degradation of hemicellulose and cellulose occurs. In addition, it is suggested that the reactions involved are complex and more energy is required to carry out these reactions [47]. However, when the degree of conversion is from 0.75 to 0.9, the pre-exponential factor decreases its value in the range of 10^1^ to 10^2^ min^−1^, which indicates that the biomass conversion is in the passive pyrolytic stage and less energy is required to continue the pyrolysis.

#### 2.3.3. Reaction Model f(α)

The master plot method was used to determine the most probable reaction model during the pyrolysis of BCS. Figure 5a shows all the theoretical models described in Appendix A and the experimental curve. The master plots were calculated using Equation (15) in the range of 0.1 to 0.9, and similarly, the experimental data were performed using the heating rate of 15 °C/min and the E_A_ determined by the SK method. As shown in Figure 5a, the experimental curves were fitted to three reaction models as a function of the degree of conversion. In this case, for a heating rate of 15 °C/min, a seventh-order reaction model is followed when α ≤ 0.25, f(α) = (1 − α)^7^ with SSE of 0.0004, when 0.3 ≤ α ≤ 0.45 follows a fourth-order model, f(α) = (1 − α)^4^ with SSE of 0.0103 and when 0.5 ≤ α ≤ 0.7 the best fit was obtained by the order 2.48 model, f(α) = (1 − α)^2.48^ with SSE of 0.0056 (Figure 5b). Here, it is evident that the order of the reaction decreases as the degree of conversion increases. This behavior can be explained by presumably the increasing probability of molecular collision caused by the change in temperature region from lower to greater decomposition of hemicellulose and cellulose, thus resulting in a decreasing reaction order from 7 to 2.48. It is important to note that the lignin content present in BCS implies a complex and rigid cross-linked phenolic structure. Therefore, when the degree of conversion is α ≥ 0.75, the reaction model is associated with combined effects of various models such as nucleation, diffusion, and generally power effects [18]. This presumption agrees with the isoconversion lines when α ≥ 0.75, as shown in Appendix A, where this deviation correlates with complex reactions [6]. It is important to highlight that a high reaction order implies a low probability of collision, and this behavior has been found in bamboo waste and cocoa shell [18,48], so presumably, given the compositional characteristic of the biomass under study, the high lignin content implies a complex and rigid cross-linked phenolic structure. Therefore, pyrolysis at a low degree of conversion is associated with the combined effects of various mechanisms such as nucleation, diffusion, and power. Hence, the high reaction order found in this study may be due to a combined effect of these mechanisms [49]. Once the values of the kinetic triplets of BCS pyrolysis are estimated, it is possible to write a kinetic expression using Equation (6) and the averages of the SK apparent activation energy, as well as the pre-exponential factors for the appropriate conversion degree range, as described in the following equations:(1)$$\frac{d\alpha }{dT}=\{\begin{array}{c} 1.5x10^{12}exp\;\left(\frac{-17335.1}{T}\right)\;{\left(1\;-\;\alpha \right)}^{7}\;,\;\alpha \;\leq 0.25\;\left(264.4-302.8\;{}^{\circ}C\right)\\ 7.5x10^{13}exp\;\left(\frac{-19082.2}{T}\right){\left(1\;-\;\alpha \right)}^{4}\;,0.3\leq \;\alpha \;\leq 0.45\;\left(316.2-343\;{}^{\circ}C\right)\\ 2.3x10^{14}\;exp\;\left(\frac{-17665.7}{T}\right)\;{\left(1\;-\;\alpha \right)}^{2.48}\;,\;0.5\leq \;\alpha \;\leq 0.7\;\left(347.5-385.6\;{}^{\circ}C\right) \end{array}\;$$

### 2.4. Thermodynamics Analysis

The thermodynamic parameters, enthalpy, Gibbs free energy, and entropy were plotted as a function of the degree of conversion (α) (Figure 6), and the numerical values are shown in Table 4. All these calculations were performed using the apparent activation energy calculated from the SK method and the pre-exponential factor using the Kissinger method at 15 °C/min. The evaluation of thermodynamic properties is essential to define the operating conditions of a process, and in this sense, the ΔH is defined as the energy used in the conversion of biomass into various pyrolytic products. The average value obtained was 117.4 kJ/mol. In addition, the difference between E_A_ and ΔH is the energy between the reactant and the activated complex. It is called the “potential energy barrier” and reveals the ease of formation of the activated complex, where the value (5.2 kJ/mol) found is less than 5.5 kJ/mol. This value indicates a feasible reaction to produce bioenergy from BCS pyrolysis, as little additional energy is required to achieve product generation, such as those reported by Mumbach et al. and Sangaré et al. [18,22]. Therefore, this indicates that favorable conditions exist for the formation of activated complexes and ease of pyrolysis reaction [18,50], i.e., pyrolysis can be carried out by providing this minimal additional energy difference [6].

The analysis and knowledge of these values are fundamental for the design of a pyrolysis reactor since the design involves adequate heat transfer to ensure that the necessary energy is provided for biomass conversion [51]. Gibbs free energy is a measure of the total energy rise during a reaction in a thermal process and provides valuable information on the spontaneity and direction of reactions. A positive value of ΔG is an indication of non-spontaneity, while a large value suggests that the feasibility of the reaction is low [49]. The average value of ΔG was 182.8 kJ/mol, with a maximum deviation of ±3.5 kJ/mol. The positive value of ΔG means that the pyrolysis reaction of BCS is not spontaneous; thus, the pyrolysis process requires additional energy for the conversion of biomass into various pyrolytic products. The ΔS values are used to qualitatively estimate the reactivity of the system [52]; in this research, an average value of −104.2 J/mol K was obtained. The negative value of ΔS shows that the reactions are close to their thermodynamic equilibrium and present higher thermodynamic reversibility. Therefore, there is a lower degree of disorder in the products than in the reactants [6,18]. Furthermore, this phenomenon occurs at a conversion close to α = 0.5, which correlates with the maximum cellulose peak shown in the DTG curves (Figure 2), since cellulose is highly reactive during pyrolysis [6,43]. Finally, it is important to highlight that the average apparent activation energy values obtained from this biomass make it an attractive energy source to be used as bioenergy.

## 3. Materials and Methods

### 3.1. Feedstock Preparation and Characterization

The Nanche (*Byrsonima crassifolia*, BCS) endocarp was collected in the state of Nayarit, México and was dried (100 °C for 24 h), crushed, and sieved to particle sizes using standard sieves according to ASTM E11 procedure as follows: 75 < Dp_1_ < 150 µm, 150 < Dp_2_ < 300 µm, 600 < Dp_3_ < 850 µm and 850 < Dp_4_ < 1680 µm, which correspond to Mesh Numbers of 200, 100, 30, and 20, respectively. Biomass characterization consisted of evaluating the following: (i) Elemental analysis and ash content using an elemental analyzer (Carlo Erba Ea-1110) coupled to ICP (Thermo Jarrell Ash IRIS/AP DUO ICP), (ii) Calorific value using an adiabatic bomb calorimeter (Parr-1341 Oxygen Bomb Calorimeter) following the ASTM D-2015-96 standard method, (iii) Lignin, cellulose, and hemicellulose content by gravimetric techniques described in ASTM (E1756-95, D1106-95, and D1103-60), and (iv) Moisture, volatiles, and ash content according to the procedure described in ASTM E (871-82), ASTM (872-82), and ASTM (1755-1795), respectively.

### 3.2. Thermogravimetric Analysis

The thermal degradation of the Nanche endocarp was investigated using a thermogravimetric analyzer (TA Instruments, Q500). Thermogravimetric analysis (TGA) data were collected in a temperature range between 30 and 900 °C to completely evaluate the deconvolution of DTG considering the region of decomposition of hemicellulose, cellulose, and lignin. According to the ICTAC recommendations [24,53,54], it is required to use slow heating rates to avoid self-heating/cooling and to generally minimize the adverse heat and mass transfer phenomena, for which the following heating rates (β) were chosen: 5, 10, and 15 °C/min under a nitrogen atmosphere (N_2_, flow rate: 100 cm^3^/min) and using a BCS mass of ~30 mg. The differential thermogravimetric analysis (DTG) was calculated using Origin Plot Software from the different data generated by the decomposition heating rates previously performed. To find the appropriate particle size (D_p_) for the kinetic analysis, samples with different particle sizes were analyzed: 200, 100, 30, and 20 mesh at 5 °C/min.

### 3.3. Kinetic Triplet Estimation

The thermochemical conversion of biomass is complex, as it involves the degradation of hemicellulose, cellulose, and lignin, as well as their interactions with and between degradation products [19,25]. For simplicity, the kinetic analysis correlates to the overall biomass pyrolysis reaction using the following overall reaction scheme:$$\mathrm{Lignocellulosic}\mathrm{biomass}\overset{k\left(T\right)}{\to }\mathrm{Vollatiles}\;+\;\mathrm{biochar}$$

The general expression for biomass pyrolysis kinetics is evaluated by the degradation rate, which is a function of temperature (T) and degree of conversion (α) and can be expressed by Equation (2).
(2)$$\frac{d\alpha }{dt}=k\left(T\right)*f\left(\alpha \right)$$
where k corresponds to the apparent rate constant, f(α) is the model-based function of conversion, and α is the degree of conversion which is expressed as:(3)$$\alpha =\frac{\left(m_{o}-m_{t}\right)}{\left(m_{o}-m_{\infty }\right)}$$
where m_o_ is the initial weight of biomass sample, m_t_ is its weight at time t, and m_ꝏ_ is the weight of char at the end of the thermal degradation. Incorporating the Arrhenius type equation, Equation (1) converts to:(4)$$\frac{d\alpha }{dt}=A\;exp\;\left(\frac{-E_{A}}{RT}\right)\;f\left(\alpha \right)$$

Here, E_A_ is the apparent activation energy (kJ/mol), A is the pre-exponential factor (min^−1^), T is the temperature (K), and R is the universal gas constant (8.314 J/K mol). During pyrolysis, the temperature increases with time since a heating rate in an inert atmosphere is fixed (β, °C/min), which can also be expressed as follows using the chain rule:(5)$$\beta =\frac{dT}{dt}=\frac{dT}{d\alpha }*\frac{d\alpha }{dt}$$

Combining Equations (4) and (5), results:(6)$$\frac{d\alpha }{dT}=\frac{A}{\beta }\;exp\left(\frac{-E_{A}}{RT}\right)\;f\left(\alpha \right)$$

Equation (6) refers to the biomass degradation during pyrolysis and is used to calculate the kinetic parameters involved in the process. The integration as a function of temperature of Equation (6) gives the following results:(7)$$g\left(\alpha \right)=\int_{0}^{\alpha }\frac{d\alpha }{f\left(\alpha \right)}=\frac{A}{\beta }\;\int_{T_{o}}^{T}exp\left(\frac{-E_{A}}{RT}\right)dT\;$$

The integral form of Equation (7) is obtained by solving the integral as follows:(8)$$g\left(\alpha \right)=\frac{AE_{A}}{\beta R}\;\int_{0}^{\infty }exp\left(-u\right)du=\frac{A\;E_{A}}{\beta \;R}p\left(x\right)$$
where g(α) is the integrated reaction model, x is equal to E_A_/RT, p(x) is the exponential integral of temperature and has no analytical solution. Generally, Equations (7) and (8) are used to determine the kinetic triplet; apparent activation energy, pre-exponential factor, and reaction model [22].

#### 3.3.1. Estimation of Apparent Activation Energy (E_A_)

Different kinetic models have been proposed to solve Equation (7) by using appropriate mathematical approximations, which are used in isoconversional methods [5,27]. In this study, for the evaluation of the kinetic triplet, the apparent activation energy (E_A_) was estimated using integral isoconversional models: Kissinger–Akahira–Sunose (KAS), Flynn–Wall–Ozawa (FWO) and Starink (SK). Which are presented below:

*Kissinger–Akahira–Sunose (KAS)*: The KAS method is an integral isoconversional method proposed by Kissinger [55] and Akahira and Sunose [56]. The KAS method uses the simple approximation for the exponential integral: px = x^−2^e^x^, introducing this approximation in Equation (7) gives the following:(9)$$ln\;\left(\frac{\beta_{i}}{T_{\alpha ,i}{}^{2}}\right)=ln\left(\frac{A_{\alpha }R}{E_{A}g\left(\alpha \right)}\right)-\frac{E_{A}}{RT_{\alpha ,i}}$$

For each degree of conversion, the $ln\left(\frac{\beta_{i}}{T_{\alpha ,i}{}^{2}}\right)\;\mathrm{vs}.\;\frac{1}{T_{\alpha ,i}}$ is plotted using the experimental data, and from the slope of the straight lines, the apparent activation energy is obtained.

*Flynn–Wall–Ozawa (FWO)*: The FWO method is an integrated isoconversional technique for the calculation of the apparent activation energy proposed by Ozawa [57] and Flynn and Wall [58], who used Doyle’s linear approximation instead of the temperature integral for Equation (7): ln P(x)≈−5.331−1−052x. The FWO method is expressed as follows:(10)$$ln\;\left(\beta_{i}\right)=ln\left(\frac{A_{\alpha }E_{A}}{g\left(\alpha \right)R}\right)-5.331-1.052\frac{E_{A}}{RT_{\alpha ,i}}$$

The apparent activation energy can be calculated through the slope of the plot between $ln\left(\beta_{i}\right)\;\mathrm{vs}.\;1/T_{\alpha ,i}$.

*Starink (SK)*: The apparent activation energy can be obtained by the method of Starink [32,59], using the following expression:(11)$$ln\;\left(\frac{\beta_{i}}{T_{\alpha ,i}{}^{1.92}}\right)=Constant-1.0008\frac{E_{A}}{RT_{\alpha ,i}}$$

Here, $ln\;\left(\frac{\beta_{i}}{T_{\alpha ,i}{}^{1.92}}\right)$ versus $\frac{1}{T_{\alpha ,i}}$ is plotted to obtain a set of straight lines at different conversions, where the apparent activation energy can be obtained from the slopes of these lines.

#### 3.3.2. Estimation of the Pre-Exponential Factor (A_α_)

The isoconversional methods detailed above are used to determine the apparent activation energy. However, the value of A_α_ calculated by these methods is considered not reliable [34]. Therefore, the standard ASTM E698-18 method based on Kissinger’s equation, described by the equation below, is used.
(12)$$ln\;\left(\frac{\beta }{T_{p}{}^{2}}\right)=ln\left(\frac{A_{\alpha }R}{E_{A}g\left(\alpha \right)}\right)-\frac{E_{A}}{RT_{p}}$$
where Tp is the temperature of the maximum peak of the DTG plot for a value of β, the value of E_A_ in this method is calculated from the data obtained by the isoconversional methods described above. That is, once the values of E_A_ for different conversion levels α are known, Equation (13) obtained from Equation (12) is used to calculate A_α_ [6,34].
(13)$$A_{\alpha }=\frac{\beta E_{A,\;\alpha_{i}}exp\left(\frac{E_{A,\;\alpha_{i}}}{RT_{p}}\right)}{RT_{p}{}^{2}}$$

#### 3.3.3. Estimation of the Reaction Model

Different kinetic methods have been detailed in the literature, and various reaction models have been proposed to describe the overall solid-state reaction [20,22]. The master plots method [26,27] uses the Taylor series approach to evaluate the exponential temperature integral and can be expressed as follows:(14)$$g\left(\alpha \right)=\frac{AE_{A}}{\beta R}p\left(x\right)=\frac{AE_{A}}{\beta R}\left[\frac{e^{-x}}{x^{2}}\left(1-\frac{2!}{x}+\frac{3!}{x^{2}}+\frac{4!}{x^{3}}+\cdots \right)\right]$$

For 20 ≤ x ≤ 50 and using the normalized equation Equation (15), masters plots are developed, considering the integral models described in Appendix A, and by plotting the experimental and model data in parallel, the reaction model associated with the thermal degradation of biomass can be defined [5].
(15)$$\lambda \left(\alpha \right)=\frac{g{\left(\alpha \right)}_{i}}{g{\left(\alpha \right)}_{0.5}}=\frac{p{\left(x\right)}_{i}}{p{\left(x\right)}_{0.5}}$$
where g(α)0.5 is the integral model at α = 0.5 and p(x)0.5 is the approximation for α = 0.5, where x = E_A,α:0.5_/RT_α:0.5_. The main reaction model is chosen by comparing the theoretical and experimental curves, where the theoretical curve with the smallest value of sums square error (SSE, Equation (16)) is selected as the reaction model [20].
(16)$$SS_{E}=\sum_{i=1}^{N}{\left(\lambda {\left(\alpha \right)}_{exp}-\lambda {\left(\alpha \right)}_{calc}\right)}^{2}$$
where N is the number of data points; λ(α)exp is the experimental master plot equation; λ(α)calc is the calculated master plot normalized equation by the integral models.

### 3.4. Estimation of Thermodynamic Parameters

Thermodynamic properties such as Gibbs free energy (∆G), enthalpy (∆H), and entropy (∆S) are a requirement to carry out a larger-scale thermochemical conversion process in terms of energy calculations [13]. These properties can be estimated using the apparent activation energy and the pre-exponential factor determined by the methods described above [6] and are calculated by the following equations:(17)$$\Delta G=E_{A}+RT_{p}\;ln\left(\frac{k_{B}\;T_{p}}{h\;A_{\alpha }}\right)$$
(18)$$\Delta H=E_{A}+RT_{p}$$
(19)$$\Delta S=\frac{\Delta H-\Delta G}{T_{p}}$$

Here k_B_ is the Boltzmann constant (1.381 × 10^−23^ J/K, h is the Plank constant 6.626 × 10^−34^ J/s. All kinetic triplet calculations and thermodynamic parameters were determined using Microsoft Excel.

## 4. Conclusions

In this study, the pyrolysis kinetics of Nanche stone (BCS) was investigated for the first time by thermogravimetric analysis. The low moisture and ash content, as well as its high volatile matter content (≥70%), make BCS a potential candidate for obtaining various bioenergy products.

The average apparent activation energies obtained from different methods (KAS, FWO, and SK) were similar to each other (~123.8 kJ/mol), and the pre-exponential factor as a function of the degree of conversion from the Kissinger method ranges from 10^5^ to 10^14^ min^−1^ for the highest pyrolytic activity stage, indicating a highly temperature reactive system and mainly the reaction model that best fits the pyrolysis of BCS corresponds to the n-order reaction models. The thermodynamic parameters revealed that there is a small difference between E_A_ and ∆H (5.2 kJ/mol), which favors the pyrolysis reaction and indicates the feasibility of the energetic process.

It is important to emphasize that the high lignin and cellulose content of BCS implies a complex and rigid cross-linked structure, and based on the results obtained from the kinetic triplet analysis (apparent activation energy, pre-exponential factor, and reaction model) revealed that the reaction pattern of this biomass is closely related to a high-order pyrolysis reaction, which presumably is the result of the combined effect of diffusion and power law models. Finally, it is important to highlight that the present study is the first report on the pyrolysis kinetics of Nanche (*Byrsonima crassifolia*) stone as biomass waste, and according to the results obtained from the kinetics triplets and thermodynamic parameters, this biomass can be considered with a high potential to be exploited as a valuable bioenergy resource.

## Figures and Tables

**Figure 1 molecules-28-00544-f001:**
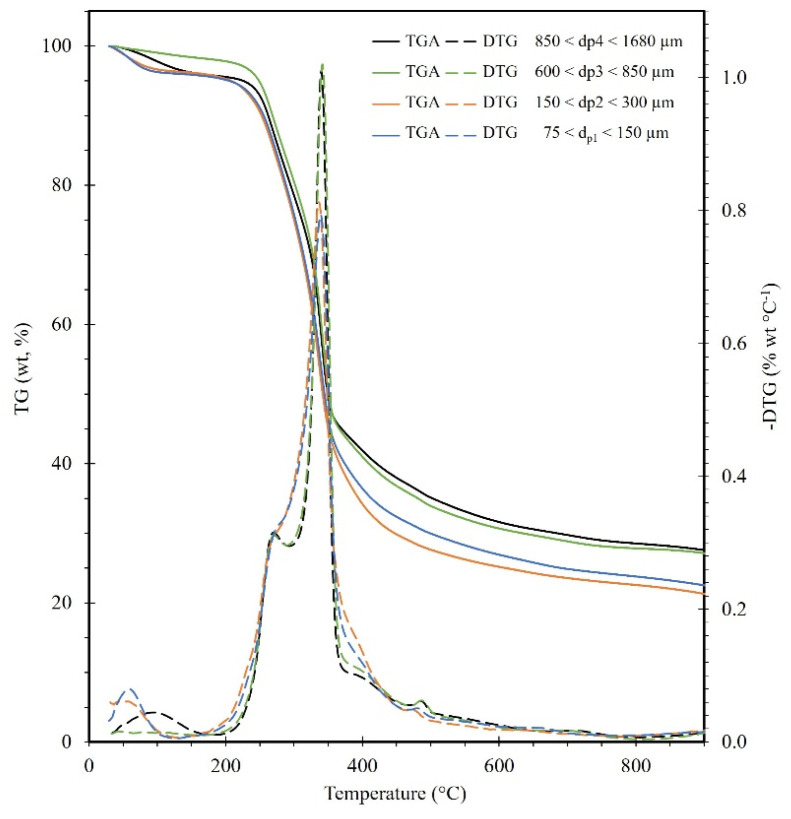
TGA and DTG curves for the pyrolysis of BCS at 5 °C/min and different particle sizes.

**Figure 2 molecules-28-00544-f002:**
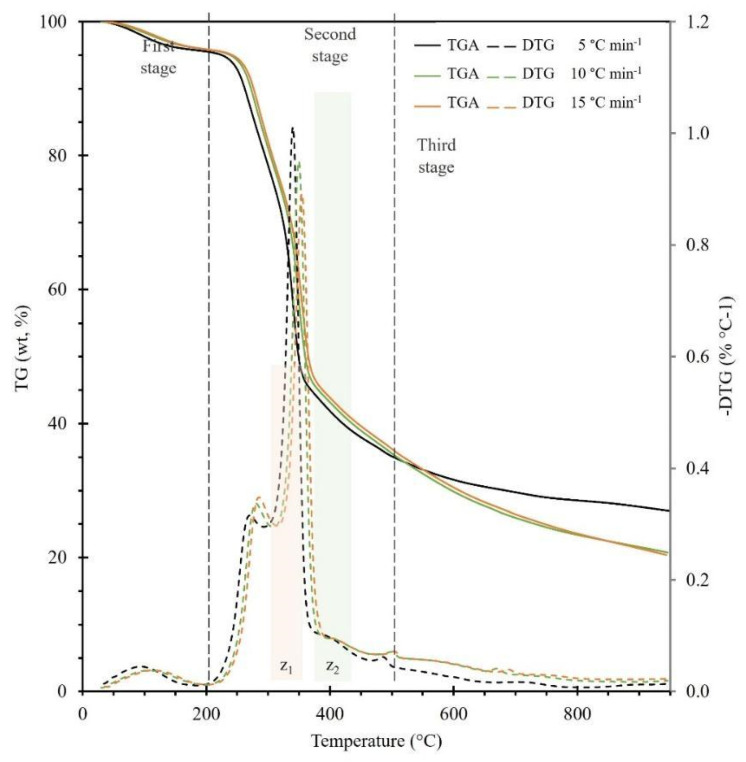
TGA and DTG curves for the pyrolysis of BCS at different heating rates 5, 10, and 15 °C/min and particle size D_p4_.

**Figure 3 molecules-28-00544-f003:**
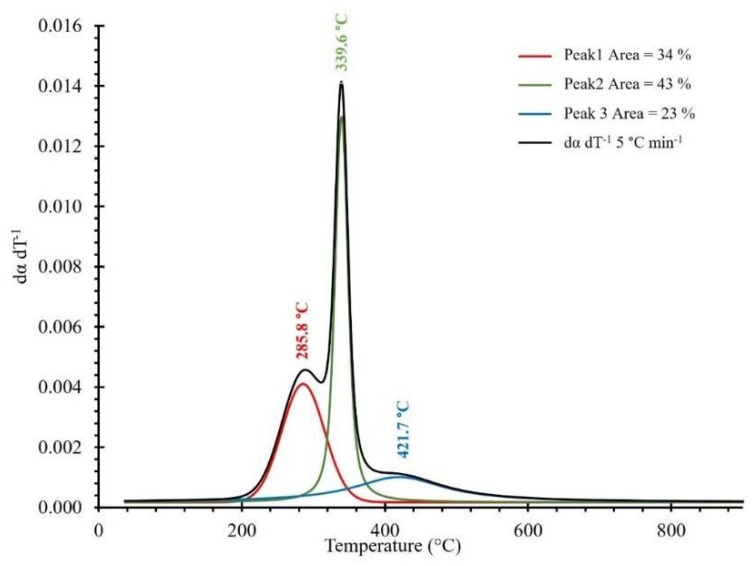
Deconvolution curves for the pyrolysis of BCS at 5 °C/min and particle size Dp_4_.

**Figure 4 molecules-28-00544-f004:**
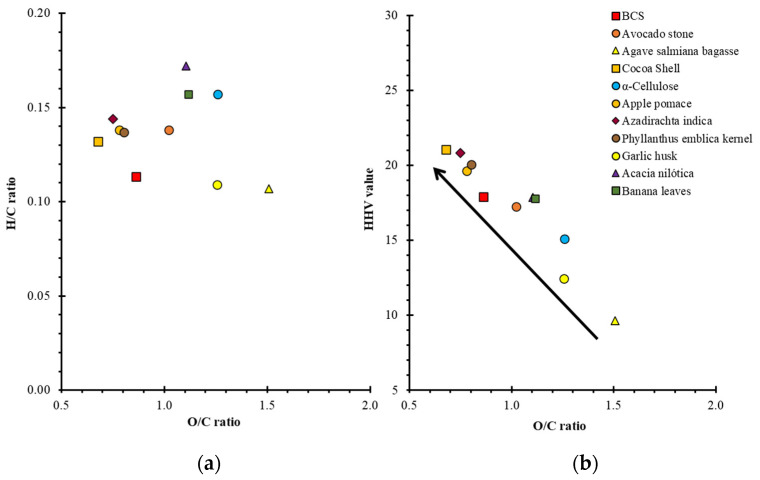
(**a**) Van-Krevelen plot and (**b**) Correlation O/C ratio vs. HHV value of biomasses.

**Figure 5 molecules-28-00544-f005:**
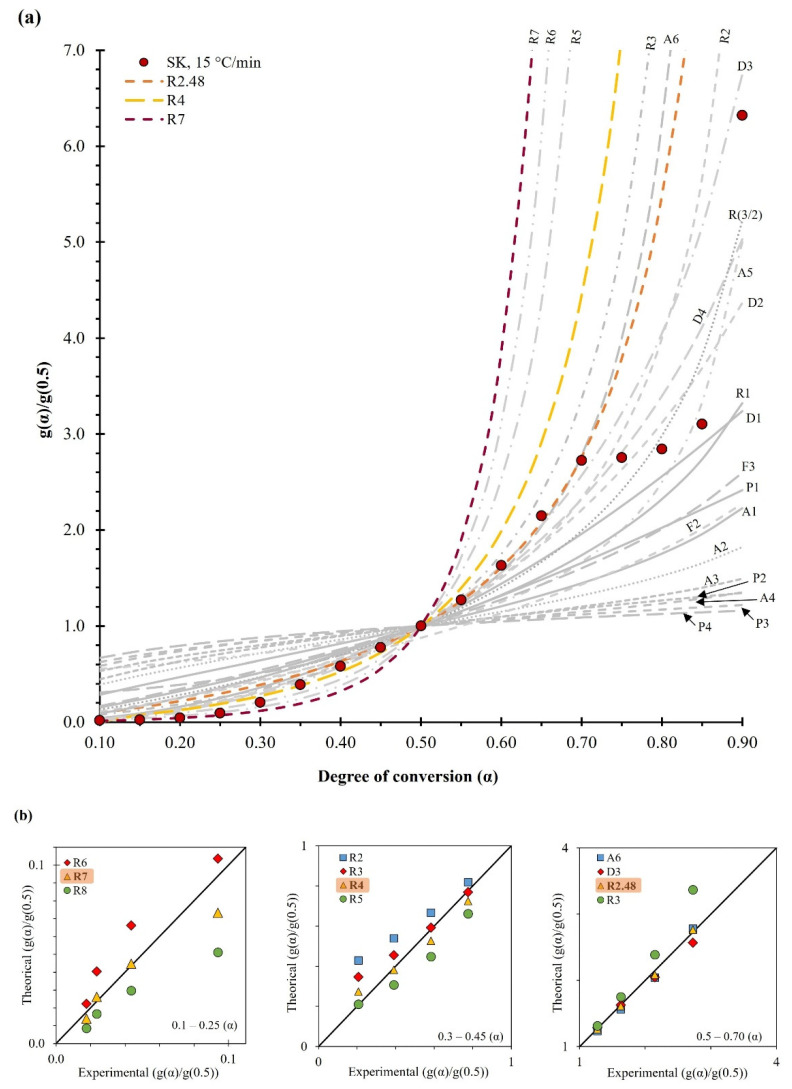
(**a**) Reaction models using generalized master plot method (g(α)/g(0.5) vs. α), and (**b**) experimental and theoretical curves (g(α)/g(0.5)).

**Figure 6 molecules-28-00544-f006:**
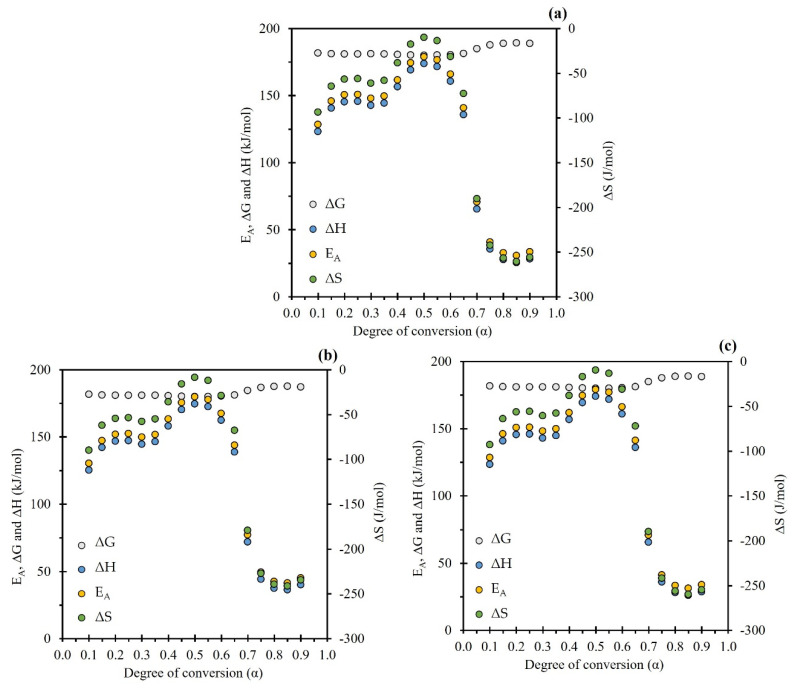
Thermodynamic parameters from the (**a**) KAS, (**b**) FWO, and (**c**) Starink methods.

**Table 1 molecules-28-00544-t001:** Characterization of BCS.

*Elemental analysis (wt %, dry basis)*	
C	49.88
H	5.65
N	1.52
O ^a^	42.95
S	N.D.
*Compositional analysis (wt %, dry extraibles-free basis)*	
Cellulose ^a^	44.16
Hemicellulose	21.17
Lignin	34.67
Extractibles with ethanol	2.01
*Proximate analysis (wt %, dry basis)*	
MC	11.3
VM	71.7
FC ^a^	14.8
Ash	2.2
*Ash composition (wt %)*	
B	0.23
Ba	0.42
Ca	6.66
Co	0.02
Cr	0.21
Cu	0.34
Fe	1.41
K	76.38
Mg	1.59
Mn	0.61
Mo	0.03
Na	7.95
Ni	0.26
Si	2.49
Sr	0.83
Ti	0.16
Zn	0.40
Physical properties (dry basis)	
Bulk density (kg/m^3^)	980
Calorific Power (MJ/kg)	14.93

MC: moisture content; VM: volatile matter; FC: fixed carbon; ^a^; by difference; N.D: non detected.

**Table 2 molecules-28-00544-t002:** Peak temperatures and residues of different particle sizes of BCS sample and different heating rates.

BCS sample (5 °C/min)	D_p1_	D_p2_	D_p3_	D_p4_
Particle size (µm)	75–150	150–300	600–850	850–1680
Peak_max._Temp. (°C)	338.85	336.28	341.49	340.03
DTG_max_ (%/°C)	0.7932	0.8125	1.020	1.01
Residue_940°C_ (%)	21.87	20.46	26.45	27.08
BCS D_p4_|Heating rate °C/min)	5 °C/min	10 °C/min	15 °C/min	
Peak_max,1_ Temp. (°C)	269.98	280.55	285.45	
Peak_max,2_ Temp. (°C)	340.03	394.56	354.54	
DTG_max_ (%/°C)	1.01	0.95	0.89	
Residue_940°C_ (%)	27.08	20.86	20.48	

**Table 3 molecules-28-00544-t003:** Comparison of apparent activation energy, structural composition, elemental ratios, and HHV values by different biomass.

Sample	E_A_ (kJ/mol)	HE ^a^ (%)	CE ^b^ (%)	LI ^c^ (%)	O/C Ratio	H/C Ratio	HHV ^d^ (MJ/kg)	Reference
BCS	122.7 ^e^	21.17	44.16	34.67	0.861	0.113	17.92	This study
Avocado stone	88.9 ^f^	76.4	3.0	17.0	1.021	0.138	17.24	Sangaré et al. [6]
*Agave salmiana* bagasse	111.6 ^f^	43.8	40.7	14.2	1.507	0.107	9.65	
Cocoa Shell	197.7 ^f^	45.4	7.8	21.5	0.678	0.132	21.06	
α-Cellulose	166.4 ^f^	-	100	-	1.258	0.157	15.10	
Apple pomace	194.8 ^f^	27.77	47.49	24.72	0.780	0.138	19.66	Baray-Guerrero et al. [19]
*Azadirachta indica*	193.7 ^f^	24.64	38.61	12.89	0.750	0.144	20.83	Mishra et al. [5]
*Phyllanthus emblica* kernel	195.1 ^f^	21.43	46.11	10.22	0.803	0.137	20.08	
Garlic husk	154.9 ^f^	29.34	41.32	17.14	1.255	0.109	12.46	Singh et al. [20]
*Acacia nilótica*	221.6 ^f^	28.64	41.66	24.20	1.105	0.172	17.86	Singh et al. [4]
Banana leaves	84 ^f^	34.34	43.34	15	1.116	0.157	17.80	Singh et al. [44]

^a^ Hemicellulose, ^b^ Cellulose, ^c^ Lignin, ^d^ Higher heating value calculated by Demirbaş formula [45], ^e^ Calculated by Starink model. ^f^ Calculated by FWO model.

**Table 4 molecules-28-00544-t004:** Apparent activation energy (E_A_) by the Starink model, pre-exponential factor (A_α_) by the Kissinger method, and thermodynamics properties for pyrolysis of BCS.

α	E_A_ (kJ/mol)	A_α_ (min^−1^)	Gibbs Free Energy ΔG (kJ/mol):	Enthalpy ΔH (kJ/mol):	Entropy ΔS (J/mol):
0.10	128.6	2.9 × 10^10^	181.8	123.3	−93.1
0.15	146.1	9.6 × 10^11^	181.1	140.9	−64.1
0.20	150.8	2.5 × 10^12^	180.9	145.6	−56.3
0.25	151.1	2.6 × 10^12^	180.9	145.9	−55.9
0.30	148.1	1.4 × 10^12^	181.0	142.9	−60.7
0.35	149.9	2.1 × 10^12^	181.0	144.7	−57.8
0.40	161.9	2.2 × 10^13^	180.6	156.7	−38.0
0.45	174.6	2.7 × 10^14^	180.2	169.4	−17.1
0.50	179.2	6.7 × 10^14^	180.0	174.0	−9.6
0.55	177.0	4.3 × 10^14^	180.1	171.7	−13.3
0.60	166.1	5.1 × 10^13^	180.4	160.9	−31.1
0.65	141.2	3.6 × 10^11^	181.3	136.0	−72.2
0.70	70.9	2.6 × 10^5^	184.9	65.7	−189.8
0.75	41.2	5.1 × 10^2^	187.7	36.0	−241.7
0.80	33.3	9.0 × 10^1^	188.8	28.1	−256.1
0.85	31.2	5.7 × 10^1^	189.1	26.0	−259.9
0.90	34.0	1.0 × 10^2^	188.7	28.8	−254.8
Average	122.7	8.6 × 10^13^	182.8	117.4	−104.2

## Data Availability

Not applicable.

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
