# Peer review of "Pyrolysis Kinetics of Byrsonima crassifolia Stone as Agro-Industrial Waste through Isoconversional Models"

_molecules, 2023, doi:10.3390/molecules28020544_

Round 1

Reviewer 1 Report

Generally, the introduction is clear with sufficient research background and the motivation of the study is understandable. The second section should be label as 2.0 Method. There should be a separation between the method with the result and discussion section. The work still lacks in clarification on the methods and some results explanation parts. The results needs to be discussed in detail and provide sufficient comparison and literature support, as this is the first work on such feedstock.

Abstract:

1.       Check on pre-exponential value

2.       Why does a small difference between EA and H favors pyrolysis?

3.       What does decreasing reaction order means? Is there no specific reaction model suitable? Or it’s a multi-step reaction?

1.0 Introduction:

4.       Reference for Mishra & Mohanty missing (Line 87)

5.       There is no literature background on the reaction model?

2.0

6.       The methods and equipment used for analysis needs to be stated for experiment replication purposes

7.       The Ash Composition (%) – Table 1 is used incorrectly. It needs to be change

8.       The values in table 1 is it measured on a dry-basis / ash-dry-basis / normal condition?

9.       How did the TGA is being conducted, why did the author choose the heating rates and temperature regions, these needs to be justify.

10.   How do you define ‘consistent behaviour’ (Line 152)?

11.   There might be a flaw in data for Fig 2. Technically, a higher heating rate will lead to a higher DTG value. The first peak make sense, but the second peak doesn’t. The explanation given in line 173 is does not justify the occurrence.

12.   Figure 6 is not being presented clearly, they should be separated.

3.0

13.   Check for proper use of subscript & superscript (Line 396)

14.   How did the samples of different size being prepared. Does the value (line 400) corresponds to mesh size?

15.   Caption of Table 5, having a different font size

4.0

16.   There is no mention of all kinetic triplets in the conclusion

17.   Author contribution part is not correctly presented.

Author Response

Title: Pyrolysis Kinetics of Byrsonima crassifolia Stone as Agro-Industrial Waste Through Isoconversional Models

Molecules

Ref. No.  2098361

Reviewer #1

Question:

  1. Check on pre-exponential value

Authors Answer:

Thank you for your careful review.

Action:

The values of the pre-exponential factor were modified to 105 to 1014 min-1 and all the subscripts in the manuscript were carefully reviewed in corrected.

Question:

  1. Why does a small difference between EA and ΔH favors pyrolysis?

Authors Answer:

Thank you for pointing out this issue.

Action:

The following paragraph was added to explain the referred behavior on lines 506-510 of the updated manuscript: “…the average value obtained was 117.4 kJ/mol, in addition, the difference between EA and ΔH is the energy between the reactant and the activated complex, it is called “potential energy barrier” and reveals the ease of formation of the activated complex, where the value (5.2 kJ/mol) found are less than 5.5 kJ/mol, this value indicates a feasible reaction to produce bioenergy from BCS pyrolysis, as little additional energy is required to achieve product generation, like those reported by Mumbach et al., and Sangaré et al. [18, 19]…”

  1. What does decreasing reaction order means? Is there no specific reaction model suitable? Or it’s a multi-step reaction?

Authors Answer:

Thank you for your comments.

Given the compositional complexity of a biomass, a large series of reactions occur during its pyrolysis. Hence, it is often possible to obtain a single model fit, such as a second-order reaction model for the Avocado Stone, whereas it is also possible to fit a combined model such as a 2.5-order reaction and nucleation for Cocoa shells [19]. Therefore, it is frequently not possible to obtain a general reaction model for the entire pyrolysis of a biomass.

Action:

The following paragraph was added in lines: 487-495.

“… It is important to highlight that a high reaction order implies a low probability of collision and this behavior has been found in bamboo waste and cocoa shell [57, 18], so presumably, given the compositional characteristic of the biomass under study, the high lignin content implies a complex and rigid cross-linked phenolic structure, therefore, pyrolysis at a low degree of conversion is associated with combined effects of various mechanisms such as nucleation, diffusion and power, hence, the high reaction order found in this study may be due to a combined effect of these mechanisms [44]…”

Regarding the issue of the meaning of decreasing reaction order the following paragraph was added in lines 481-485:

“…Here, it is evident that the order of reaction is decreasing as the degree of conversion increases. This behavior can be explained by presumably there is increasing probability of molecular collision caused by the change in temperature region from lower to greater decomposition of hemicellulose and cellulose, thus resulting in a decreasing reaction order from 7 to 2.48…”

  1. Reference for Mishra & Mohanty missing (Line 87)

Authors Answer:

Thank you for pointing out this mishap.

The reviewer's suggestion was followed, and the missing reference was included in the revised version of the manuscript.

  1. There is no literature background on the reaction model?

Authors Answer:

Thank you for your comments.

Two references were added in lines 80-81, in which the use of the reaction models studied in this manuscript is also detailed and the background of the use of these reaction models is further explained in the methodology (lines 229-245).

Here are the additions to the manuscript:

Introduction: (lines79-83)

“…From this analysis, it is possible to evaluate the kinetic triplet consisting in the apparent activation energy (EA), pre-exponential factor (Aα) and reaction mechanism or kinetic model, f(α) [6, 19] and in turn, estimate the thermodynamic properties of the process (∆H, ∆G and ∆S). These parameters are essential to investigate and evaluate the energetic feasibility of a pyrolysis process [13, 18]….”

Section 2.3.3 (lines 229-244)

“…2.3.3 Estimation of the reaction model

Different kinetic methods have been detailed in the literature and various reaction models have been proposed to describe the overall solid-state reaction [19, 21]. The master plots method [25], uses the Taylor series approach to evaluate the exponential temperature integral and can be expressed as follows:

                    (13)

For 20 ≤ x ≤ 50 and using the normalized equation [Eq. (14)], masters plots are developed, considering the integral models described in Table S1 in the Supplementary Information, and by plotting the experimental and model data in parallel, the reaction model associated with the thermal degradation of biomass can be defined [5].

                                               (14)

Where g(α)0.5 is the integral model at α = 0.5 and p(x)0.5 is the approximation for α = 0.5, where x = EA,α:0.5/RTα:0.5. The main reaction model is chosen by comparing the theoretical and experimental curves, where the theoretical curve with the smallest value of sums square error (SSE, Eq. 15) is selected as the reaction model [21]…”

  1. The methods and equipment used for analysis needs to be stated for experiment replication purposes

Thank you for pointing out this issue:

Action:

Section 2.0 was added and called “Materials and Methods” where all the methods and equipment employed in the research were included. Please see new section 2.0 in the revised version of the manuscript.

  1. The Ash Composition (%) – Table 1 is used incorrectly. It needs to be change

Thank you for your careful review of this Table:

Action:

Table 1 was modified according to suggestions of the reviewer by adding the wt %, dry basis considering ASTM 1755-95 procedure.

  1. The values in table 1 is it measured on a dry-basis / ash-dry-basis / normal condition?

Thank you for your careful review of this Table:

Action:

Table 1 was modified by adding “dry-basis” conditions where appropriate.

  1. How did the TGA is being conducted, why did the author choose the heating rates and temperature regions, these needs to be justify.

Thank you for comments:

Action:

TGA methodology is now described in section 2.2

Heating rates and temperature regions were chosen because:

According to ICTAC recommendations [55, 56] it is required to use heating rates (< 20 °C/min) to avoid self-heating/cooling and in general minimize mass and heat transfer phenomena.

Action:

A short discussion was added in lines: 131-133. Since lignin ends its decomposition at   ̴700 °C. Therefore, it is important to evaluate a wider temperature region to be able compare the deconvolution analysis of the DTG´s with the compositional content of the BCS biomass. If the section would have been terminated at a temperature below 700 °C obtained data, it would not provide a correct estimation of the deconvolution analysis.

A short discussion was added in lines: 127-130.

Lines 127-133

“…The thermal degradation of Nanche endocarp was investigated using a thermogravimetric analyzer (TA Instruments, Q500). Thermogravimetric analysis (TGA) data were collected in a temperature range between 30 and 900 °C, to completely evaluate the deconvolution of DTG considering the region of decomposition of hemicellulose, cellulose, and lignin. According to the ICTAC recommendations [55, 56], it is required to use slow heating rates to avoid the self-heating/cooling and to generally minimize the adverse heat and mass transfer phenomena, for which the following heating rates (β) were chosen: 5, 10 and 15 °C/min under a nitrogen atmosphere (N2, flow rate: 100 cm3/min) and using a BCS mass of ~30 mg. The differential thermogravimetric analysis (DTG) was calculated using Origin Plot Software from the different data generated by the decomposition heating rates previously performed. To find the appropriate particle size (Dp) for the kinetic analysis, samples with different particle sizes were analyzed: 200, 100, 30 and 20 mesh at 5 °C/min.…”

  • How do you define ‘consistent behavior’ (Line 306)?

Thank you for comments:

The “consistent behavior” refers to the fact that the DTG plot (Fig. 1) of particle size Dp4 presents well defined characteristic peaks of hemicellulose and cellulose, which will facilitate the deconvolution analysis detailed in section 3.2.2

Action:

The paragraph was rewritten as follows (lines 304-308):

“…For the analysis of the effect of heating rate, the particle size (850 µm < Dp4 < 1680 µm, mesh #20) was chosen because it presents the most consistent behavior that is, this DTG curve presents well defined characteristic peaks of hemicellulose, cellulose, and lignin, which will facilitate the complete analysis by means of the deconvolution technique that is described below…”

  • There might be a flaw in data for Fig 2. Technically, a higher heating rate will lead to a higher DTG value. The first peak make sense, but the second peak doesn’t. The explanation given in line 173 is does not justify the occurrence.

Thank you for pointing out this issue:

According to Várhegyi et al., (https://doi.org/10.1016/0961-9534(95)92631-H) mass transfer problems caused by high heating rates can delay the decomposition process, and in the case of cellulose the presence of reaction products during decomposition can initiate autocatalytic reactions and cellulose can be consumed below the maximum cellulose decomposition temperatures, thus causing a shift and decrease in the maximum peak cellulose degradation of DTGs as in the present study.

Action:

The following paragraph was added to the manuscript in lines XXX to explain the behavior pointed out by the reviewer:

Lines 334-342:

“…Concerning the fact that a reverse behavior is observed the temperature range of 315-400 °C with respect to the hemicellulose decomposition this can be explained from the behavior of the cellulose decomposition reported by Várhegyi et al., [59] where mass transfer problems caused by high heating rates can delay the decomposition process, and in the case of cellulose the presence of reaction products during its decomposition can initiate autocatalytic reactions and cellulose can be consumed below the maximum cellulose decomposition temperatures, thus causing a shift and decrease in the maximum peak cellulose degradation of DTG´s as in the present study…”

  • Figure 6 is not being presented clearly; they should be separated.

Thank you for your suggestion:

Action:

Figure 6 was separated and were magnified in size. Now these are 6(a) Reaction models using generalized master plot method (g(α)/g(0.5) vs α) and 6(b) Experimental and theorical curves (g(α)/g(0.5)).

  1. Check for proper use of subscript & superscript (Line 396)

Action:

All subscripts and superscripts were reviewed and when applicable corrected throughout the entire manuscript.

  • How did the samples of different size were prepared. Does the value (line 400) correspond to mesh size?

Thank you for your input in describing the preparation of the samples.

Action:

Detailed information was added in the methodology section (line 115-118 and 138), and the nomenclature by mesh number (#Mesh) has been added in section 3.2.1 (Line 305).

Here:

Lines: 114-118:

“…The Nanche (Byrsonima crassifolia, BCS) endocarp was collected in the state of Nayarit, México and was dried (100 °C for 24 h), crushed and sieved to particle sizes using standard sieves according to ASTM E11 procedure as follows: 75 < Dp1 < 150 µm, 150 < Dp2 < 300 µm, 600 < Dp3 < 850 µm and 850 < Dp4 < 1680 µm, which correspond to Mesh Numbers of 200, 100, 30 and 20, respectively…”

Lines 135-139:

“…The differential thermogravimetric analysis (DTG) was calculated using Origin Plot Software from the different data generated by the decomposition heating rates previously performed. To find the appropriate particle size (Dp) for the kinetic analysis, samples with different particle sizes were analyzed: 200, 100, 30 and 20 mesh at 5 °C/min…”

Lines 304-306:

“…For the analysis of the effect of heating rate, the particle size (850 µm < Dp4 < 1680 µm, mesh #20) was chosen because it presents the most consistent behavior that is, this DTG curve presents…”

  • Caption of Table 5, having a different font size

Thank you for your careful review of this Table:

Action:

All the fonts were homogenized in Table 5.

  • There is no mention of all kinetic triplets in the conclusion

Thank you for your suggestion:

Action:

The conclusion was modified to mention the kinetic triplets according to the reviewer's comments.

Lines 563-568:

“…It is important to emphasize that the high lignin and cellulose content of BCS implies a complex and rigid cross-linked structure and based on the results obtained from the kinetic triplet analysis (apparent activation energy, pre-exponential factor and reaction model) revealed that the reaction pattern of this biomass is closely related to a high-order pyrolysis reaction, which presumably is the result of the combined effect of diffusion and power law models…”

  1. Author contribution part is not correctly presented.

Thank you for your comment:

Action:

All author contributions to the manuscript were enlisted:

Lines: 578-594

“…Author Contributions: For research articles with several authors, a short paragraph specifying their individual contributions must be provided. The following statements should be used “Conceptualization, A.L. (Alejandro López-Ortiz), and J.M.S. (Jonathan M. Sanchez-Silva); methodology, R. O. (Raúl Ocampo-Pérez), E.P. (Erika Padilla-Ortega) and D.S. (Diakaridia Sangaré); software, M.A.E. (Miguel A. Escobedo-Bretado), J.L.D (Jorge L. Domínguez-Arvizu) and B.C.H. (Blanca C. Hernández-Majalca); validation, V.H.C. (Virginia Hidolina Collins Martínez), J.M.S. (Jesús M. Salinas-Gutiérrez); investigation, A.L. (Alejandro López-Ortiz), J.M.S. (Jonathan M. Sanchez-Silva) and V.H.C. (Virginia Hidolina Collins Martínez); data curation, M.A.E. (Miguel A. Escobedo-Bretado), J.L.D (Jorge L. Domínguez-Arvizu) and B.C.H. (Blanca C. Hernández-Majalca); writing—original draft preparation, A.L. (Alejandro López-Ortiz), J.M.S. (Jonathan M. Sanchez-Silva); writing—review and editing, A.L. (Alejandro López-Ortiz), J.M.S. (Jonathan M. Sanchez-Silva); visualization, J.L.D (Jorge L. Domínguez-Arvizu) and B.C.H. (Blanca C. Hernández-Majalca); supervision, A.L. (Alejandro López-Ortiz) and V.H.C. (Virginia Hidolina Collins Martínez); project administration, A.L. (Alejandro López-Ortiz) and V.H.C. (Virginia Hidolina Collins Martínez). All authors have read and agreed to the published version of the manuscript…”

Reviewer 2 Report

Presented manuscript descibes the kinetic aspects of pyrolysis of Nanche stone BSC (Byrson- 16 ima crassifolia). The object is heterogeneous and complex, thus one cannot explain to see the simple kinetics here. Nevertheless, the waste removal is an important topic and it deserves studying. Overall, the manuscript needs revision to improve the level of discussion, before it can be accepted for publication at Molecules. My comments from the beginning:

title "Iso-Conversional" -> use "Isoconversional" instead, as it is more common within community

abstract "1010 to 1014 min-1" - apply the upper case for these

abstract - (5.18 kJ/mol ...126.29 kJ/mol), - given the complexity of the process you study, these parameters cannot be determined with such a high accuracy. please, provide less digits

introduction - "and differential thermal analysis (DTGA)." - should be DTA for differential thermal analysis or DTG for analysis of differential TGA data

introduction - please, discuss the recent overview of the pyrolysis kinetics field - https://doi.org/10.3390/thermo2040029

introduction - "and the ICTAC (International Confederation for Thermal Analysis and Calorimetry) committee recommends the integral isoconversional methods such as: Kissinger-Akahira-Sunose (KAS), Flynn-Wall- Ozawa (FWO) and Starink (SK), as the most valid and reliable for the estimation of apparent kinetics [5, 21]." - Refs.5,21 refer not the the discussed ICTAC recommendations, not to the original publications of the discussed methods either. Please, correct it, by giving the proper references, viz., for ICTAC recommendations the relevant to the topic: 

https://doi.org/10.1016/j.tca.2011.03.034

https://doi.org/10.1016/j.tca.2020.178597

https://doi.org/10.1016/j.tca.2022.179384

This was on the references, on the message of the above statement, it is not correct enough. In fact, of these three integral isoconversional methods (FWO, KAS, SK) only SK can be considered as accurate. What are recommended by ICTAC (and in the above refered paper, https://doi.org/10.3390/thermo2040029) are Friedman and advanced Vyazovkin methods, they eliminate the temperature integral approximations and averaging over 0..aplha regions (that present for FWO, KAS, SK).

results, Figure 3 - which mathematical form of peaks have you used for deconvolution. As for this procedure, the first paper on it is apparently https://doi.org/10.1021/jp110895z 

Figure 4 - please, put (a)-(c) plots in the Supplementary material

Section 2.3.3 - please give some explanation for an unrealistic reaction orders you suggest

Table 5 should be moved to the Supplementary material, as it contains the common information 

Author Contributions: - please provide the initials of the authors instead of X.X. here

Author Response

ANSWERS TO REVIEWERS ABOUT THE PAPER:

Title: Pyrolysis Kinetics of Byrsonima crassifolia Stone as Agro-Industrial Waste Through Isoconversional Models

Molecules

Ref. No.  2098361

Reviewer #2

  1. Title "Iso-Conversional" -> use "Isoconversional" instead, as it is more common within community.

Thank you for your suggestion:

Action:

The term isoconversional was corrected in the title and throughout the entire manuscript.

  1. Abstract "1010 to 1014 min-1" - apply the upper case for these

Thank you for your careful review.

Action:

The values of the pre-exponential factor were modified to 105 to 1014 min-1 and all the subscripts in the manuscript were carefully reviewed in corrected.

  1. Abstract - (5.18 kJ/mol ...126.29 kJ/mol), - given the complexity of the process you study, these parameters cannot be determined with such a high accuracy. please, provide less digits.

Thank you for your suggestion:

Action:

The values were modified according to the reviewer's comments.

  1. Introduction - "and differential thermal analysis (DTGA)." - should be DTA for differential thermal analysis or DTG for analysis of differential TGA data.

Thank you for your advice:

Action:

The acronym was modified according to the reviewer's comments within the entire manuscript.

  1. Introduction - please, discuss the recent overview of the pyrolysis kinetics field - https://doi.org/10.3390/thermo2040029.

Thank you for your suggestion:

Action:

The manuscript was modified according to the advice from the reviewer:

Lines: 90-96

“…The kinetic analysis of biomass thermal degradation is generally based on the degradation rate equation developed by Friedman in 1964 [23], and the ICTAC (International Confederation for Thermal Analysis and Calorimetry) committee has evaluated integral isoconversional methods such as: Kissinger-Akahira-Sunose (KAS), Flynn-Wall-Ozawa (FWO) and Starink (SK), and concluded that the SK method is the most valid and reliable integral isoconversional method for the estimation of apparent kinetics [56]…”

  1. Introduction - "and the ICTAC (International Confederation for Thermal Analysis and Calorimetry) committee recommends the integral isoconversional methods such as: Kissinger-Akahira-Sunose (KAS), Flynn-Wall- Ozawa (FWO) and Starink (SK), as the most valid and reliable for the estimation of apparent kinetics [5, 21]." - Refs.5,21 refer not the the discussed ICTAC recommendations, not to the original publications of the discussed methods either. Please, correct it, by giving the proper references, viz., for ICTAC recommendations the relevant to the topic: 

https://doi.org/10.1016/j.tca.2011.03.034

https://doi.org/10.1016/j.tca.2020.178597

https://doi.org/10.1016/j.tca.2022.179384

This was on the references, on the message of the above statement, it is not correct enough. In fact, of these three integral isoconversional methods (FWO, KAS, SK) only SK can be considered as accurate. What are recommended by ICTAC (and in the above refered paper, https://doi.org/10.3390/thermo2040029) are Friedman and advanced Vyazovkin methods, they eliminate the temperature integral approximations and averaging over 0..aplha regions (that present for FWO, KAS, SK).

Thank you so much for such an important correction!!

According to the reviewer's comments, the references used in lines 90-96 were modified, and the activation energy used for the calculations of the reaction model and thermodynamic parameters were modified to those of the Starink model, considering the reviewer's comments. Under this modification, as a result the heating rate for the calculations of the reaction model and thermodynamic parameters was changed from 10 to 15 °C/min to properly fit the Starink model as well as the corresponding reaction models. All involved figures and tables were updated considering these changes.

Action:

Figures 5 and 6 and Table S2 were modified accordingly, and the discussion related to these changes were updated.

  1. Results, Figure 3 - which mathematical form of peaks have you used for deconvolution. As for this procedure, the first paper on it is apparently https://doi.org/10.1021/jp110895z 

Thank you for your suggestion:

Action:

The mathematical form used in the deconvolution was described (Lines: 354-357), and the reference suggested by the reviewer [58] was added.

Lines: 360-364

“…. Fig. 3 shows the results obtained using this technique for the heating rate of β = 5 °C/min. A Gaussian equation was used to fit each peak as described by Perejón et al., [58] where a mathematical model was proposed, and its integration allows to estimate the percentage of hemicellulose, cellulose and lignin content corresponding to the Nanche stone…”

  1. Figure 4 - please, put (a)-(c) plots in the Supplementary material

Thank you for your suggestion:

Action:

The corresponding Figures indicated by the reviewer were placed in the supplemental section of the manuscript.

  1. Section 2.3.3 - please give some explanation for an unrealistic reaction order you suggest

Authors Answer:

Thank you for your comments.

Action:

The following paragraph was added in lines: 487-495.

“… It is important to highlight that a high reaction order implies a low probability of collision and this behavior has been found in bamboo waste and cocoa shell [57, 18], so presumably, given the compositional characteristic of the biomass under study, the high lignin content implies a complex and rigid cross-linked phenolic structure, therefore, pyrolysis at a low degree of conversion is associated with combined effects of various mechanisms such as nucleation, diffusion and power, hence, the high reaction order found in this study may be due to a combined effect of these mechanisms [44]…”

Regarding the issue of the meaning of decreasing reaction order the following paragraph was added in lines 481-485:

“…Here, it is evident that the order of reaction is decreasing as the degree of conversion increases. This behavior can be explained by presumably there is increasing probability of molecular collision caused by the change in temperature region from lower to greater decomposition of hemicellulose and cellulose, thus resulting in a decreasing reaction order from 7 to 2.48…”

  • Table 5 should be moved to the Supplementary material, as it contains the common information. 

Thank you for your suggestion:

Action:

The corresponding Table indicated by the reviewer was placed in the supplemental section of the manuscript.

  • Author Contributions: - please provide the initials of the authors instead of X.X. here

Thank you for your comment:

Action:

All author contributions to the manuscript were enlisted:

Lines: 578-594

“…Author Contributions: For research articles with several authors, a short paragraph specifying their individual contributions must be provided. The following statements should be used “Conceptualization, A.L. (Alejandro López-Ortiz), and J.M.S. (Jonathan M. Sanchez-Silva); methodology, R. O. (Raúl Ocampo-Pérez), E.P. (Erika Padilla-Ortega) and D.S. (Diakaridia Sangaré); software, M.A.E. (Miguel A. Escobedo-Bretado), J.L.D (Jorge L. Domínguez-Arvizu) and B.C.H. (Blanca C. Hernández-Majalca); validation, V.H.C. (Virginia Hidolina Collins Martínez), J.M.S. (Jesús M. Salinas-Gutiérrez); investigation, A.L. (Alejandro López-Ortiz), J.M.S. (Jonathan M. Sanchez-Silva) and V.H.C. (Virginia Hidolina Collins Martínez); data curation, M.A.E. (Miguel A. Escobedo-Bretado), J.L.D (Jorge L. Domínguez-Arvizu) and B.C.H. (Blanca C. Hernández-Majalca); writing—original draft preparation, A.L. (Alejandro López-Ortiz), J.M.S. (Jonathan M. Sanchez-Silva); writing—review and editing, A.L. (Alejandro López-Ortiz), J.M.S. (Jonathan M. Sanchez-Silva); visualization, J.L.D (Jorge L. Domínguez-Arvizu) and B.C.H. (Blanca C. Hernández-Majalca); supervision, A.L. (Alejandro López-Ortiz) and V.H.C. (Virginia Hidolina Collins Martínez); project administration, A.L. (Alejandro López-Ortiz) and V.H.C. (Virginia Hidolina Collins Martínez). All authors have read and agreed to the published version of the manuscript…”

Round 2

Reviewer 1 Report

The authors has made significant changes prior to the previous comments. There are still some minor issues to be clarify .

1.       Reply to Question 3: How did the author end up with the claim that the reaction order of 2.48 is suitable to describe a portion of the conversion? How did the author select 2.48 to be specific? Was this done by trying all reaction order model from 2.01, 2.02…… until 2.99?

2.       Reply to Question 5: As previously describe that the authors need to specify the methods in detail for replication. In this work, the reaction model used integrates the g(a)/g(0.5). There are master plot approach that used only g(a) as the model fitting approach. Why did the author select this technique and not the latter one?

3.       The explanation given at line 401 & 423 are the same. Don’t need to repeat the same explanation in the manuscript.

4.       Table 4 is not complete in the manuscript (partial of the table is missing on the left side)

5.       Cross-check for all ‘EA’ terms in the manuscript and figures used. They are not consistent. Some are presented as ‘EA’

Author Response

(2nd Round) ANSWERS TO REVIEWERS ABOUT THE PAPER:

Title: Pyrolysis Kinetics of Byrsonima crassifolia Stone as Agro-Industrial Waste Through Isoconversional Models

Molecules

Ref. No.  2098361

Reviewer #1

Question:

  1. Reply to Question 3: How did the author end up with the claim that the reaction order of 2.48 is suitable to describe a portion of the conversion? How did the author select 2.48 to be specific? Was this done by trying all reaction order model from 2.01, 2.02…… until 2.99?

Authors Answer:

We apologize for not answering your Question 3 properly.

Here is the required explanation:

The equation of the reaction model of order "n" already integrated has the following form:  this equation when incorporated into the master plot equation (, Eq. 14). It is possible to adjust the parameter "n" by a nonlinear fitting procedure (Origin 9.0 Software) between the experimental λ(α) values and the λ(α) values of the reaction model n, decreasing the sums square error (SSE, Eq. 15) between both.  Below is a table showing the SSE values for the models that gave the best fitting results in the same reaction than the 2.48 reaction order.

Reaction model

SSE

R2

0.257

R2.48

0.005

R3

0.489

A6

0.019

D3

0.056

Therefore, the one with the lowest SSE value was chosen which is the n=2.48 reaction model. This type of fitting procedure for the constant "n" has also been used by: [6].

Question:

  1. Reply to Question 5: As previously describe that the authors need to specify the methods in detail for replication. In this work, the reaction model used integrates the g(a)/g(0.5). There are master plots approach that used only g(a) as the model fitting approach. Why did the author select this technique and not the latter one

Authors Answer:

Thank you for your comments.

When only g(a) is used, it is not possible to obtain a fair good approximation between the experimental values and the kinetic models. In addition, it has been the experience of this research group [6] that by using the normalized equations by Master Plots it is possible to obtain a very good fitting approach of the various models as shown in Figure 5 of the manuscript.

The method used for the master plots consists of using the experimental data at a chosen heating rate, and converting them with the equation x = EA,α5/RTα, then the values of p(x) must be found taking into account the following equation:

Subsequently, we obtain the values of:

And the experimental λ(α) values are compared with the λ(α) values for each model evaluated.

This technique of master plot g(a)/g(0.5) was chosen because in this way the experimental and model values are better visualized, a figure is shown below, if only the term g(a) would be used to make this plot:

t

  1. The explanation given at line 401 & 423 are the same. Don’t need to repeat the same explanation in the manuscript.

Authors Answer:

Thank you for your comments.

Action:

The lines were modified: 401-406

“… According to the ICTAC recommendations, the Starink method was chosen as the best model that represents the apparent activation energy during BCS pyrolysis. These results obtained with SK model were in good agreement with a deviation of less than 6%, which validates the reliability of the calculations and confirmed the predictive power of the SK method, and the data obtained from this method was used to perform subsequent calculations in the present study…”

Furthermore, Lines 424-427 were deleted to avoid repeating the same explanation.

  1. Table 4 is not complete in the manuscript (partial of the table is missing on the left side)

Authors Answer:

Thank you for pointing out this table.

Corrections on Table 4 were made accordingly.

  1. Cross-check for all ‘EA’ terms in the manuscript and figures used. They are not consistent. Some are presented as ‘EA’

Authors Answer:

Thank you for your comments.

The EA terms in Fig. 6 were modified and corrected.

Reviewer #2

  1. The authors made good work in improving the level of discussion in acordance with referee's comments. One previous point is not addresed in revision: "According to the ICTAC recommendations [55, 56]" - this passage in text should also refer to most recent guidelines for complex kinetics analysis (https://doi.org/10.1016/j.tca.2022.179384) After it's inclusion the manuscript can be accepted for publication.

Thank you for your suggestion:

Action:

The suggested reference by the reviewer was added in line: 131-133

“… According to the ICTAC recommendations [55, 56, 60], it is required to use slow heating rates to avoid the self-heating/cooling and to generally minimize the adverse heat and mass transfer phenomena…”

Reviewer 2 Report

The authors made good work in improving the level of discussion in acordance with referee's comments. One previous point is not addresed in revision:

"According to the ICTAC recommendations [55, 56]" - this passage in text should also refer to most recent guidelines for complex kinetics analysis (https://doi.org/10.1016/j.tca.2022.179384)

After it's inclusion the manuscript can be accepted for publication

Author Response

(2nd Round) ANSWERS TO REVIEWERS ABOUT THE PAPER:

Title: Pyrolysis Kinetics of Byrsonima crassifolia Stone as Agro-Industrial Waste Through Isoconversional Models

Molecules

Reviewer #2

  1. The authors made good work in improving the level of discussion in acordance with referee's comments. One previous point is not addresed in revision: "According to the ICTAC recommendations [55, 56]" - this passage in text should also refer to most recent guidelines for complex kinetics analysis (https://doi.org/10.1016/j.tca.2022.179384) After it's inclusion the manuscript can be accepted for publication.

Thank you for your suggestion:

Action:

The suggested reference by the reviewer was added in line: 131-133

“… According to the ICTAC recommendations [55, 56, 60], it is required to use slow heating rates to avoid the self-heating/cooling and to generally minimize the adverse heat and mass transfer phenomena…”